# The influence of a quick educational video intervention on COVID-19-related knowledge in Ecuador

Marco Faytong-Haro[1,2,3], Genesis Camacho-Leon[4,5], Roberto Páez-Plúas[2,6], Azza Sarfraz[7], Zouina Sarfraz[8], Jack Michel[5], Ivan Cherrez-Ojeda[1,9]*

1 School of Health, Universidad de Especialidades Espíritu Santo, Guayas, Samborondón, Ecuador, 2 Ecuadorian Development Research Lab, Daule, Guayas, Ecuador, 3 Sociology and Demography Department, The Pennsylvania State University, State College, PA, University Park, United States of America, 4 División de Estudios para Graduados, Facultad de Medicina, Universidad del Zulia, Maracaibo, Venezuela, 5 Division of Clinical and Translational Research, Larkin Community Hospital, South Miami, FL, United States of America, 6 Francisco de Quito (USFQ), Universidad San, Quito, Ecuador, 7 Department of Pediatrics and Child Health, The Aga Khan University, Karachi, Pakistan, 8 Department of Research and Publications, Fatima Jinnah Medical University, Lahore, Pakistan, 9 Respiralab Research Center, Guayaquil, Ecuador

* ivancherrez@gmail.com

**Data Availability Statement:** All relevant data are within the paper and its Supporting Information files.

## Abstract

Background: Coronavirus disease (COVID-19) is a pandemic that has spread worldwide. Since its discovery, health measures have been put in place to help stop it from spreading. Proper education about COVID-19 is important because it helps people to follow health control measures and learn more about the disease. Objective: This study aimed to compare people´s knowledge of COVID-19 before and after a brief video-based educational intervention. Methods: 87 participants in Ecuador were recruited from a dataset of COVID-positive patients in Ecuador between December 2021 and February 2022. This was a cross-sectional, pre- and post-intervention study. First, COVID-19 knowledge was evaluated and then an educational intervention was provided as a video. After the intervention, the same knowledge questions were used to test the participants, and marginal homogeneity-based chi-square tests were employed for comparison. Results: After watching the educational video, participants knew more about the age group most likely to get the disease and their knowledge of how long it takes for Covid to spread. Their knowledge of other aspects of COVID-19 has also increased. Conclusion: This study shows that educational intervention positively affects the knowledge of people who watch it. At the end of the study, after the intervention, the study participants knew more than they had before. This could be a useful tool for identifying possible pandemics.

## Introduction

COVID-19 first appeared in December 2019 as a series of pneumonia cases of unknown etiology [1]. Since it was declared a worldwide pandemic in March 2020, most countries have

**Funding:** This research was funded by Universidad Espíritu Santo, grant number 2021-MED-001. The funders had no role in study design, data collection and analysis, decision to publish, or preparation of the manuscript.

adopted various policies to control its transmission [2]. The most common actions were public campaigns on washing one's hands as frequently as possible, utilizing a face-covering mask that protects the nose, mouth, and chin; and to avoid social agglomeration. Moreover, nations such as Brazil have also employed algorithms that predicted the appropriate time and duration of social distancing policies on their territory using them to control transmission [3–6]. By the end of 2021, more than 620 million confirmed cases of COVID-19, including more than 6.5 million deaths. With the COVID-19 vaccine creation, governments worldwide achieved to inoculate approximately twelve billion people [7].

Public health efforts globally aimed to stop COVID-19 transmission by keeping people informed and encouraging them to adhere to recommended safety measures [8]. Part of the strategy involved sharing trustworthy information with public; therefore, they become more conscious of their context, disease, and prevention. Social distancing, for example, has proven to be the primary strategy for reducing the frequency of congregation in socially dense community settings, such as schools or workplace [3]. Nonetheless, indirect approaches have been employed educational intervention to increase social awareness and societal engagement [9].

For example, telephone health education has been proven effective in improving knowledge and practices regarding COVID-19 nature and prevention [10]. Disaster education via telephone may boost civic involvement and public knowledge [11–14]. The latter is a significant variable in adherence to control measures. Without transparent and science-based messages, knowledge gaps could be created between the populations, thereby reducing the efficacy of implemented control measures [15–17]. Another study samples 295 adults aged 55 years in Hong Kong and implemented a telephone health education program for the 2003 SARS virus. As a result, the knowledge of SARS transmission improved [18]. In addition, anxiety levels decreased even when sometimes physicians fail to acknowledge or treat this disorder [19]. Therefore, telephone contact is a practical way of providing health.

When face-to-face measures are not feasible, providing education and supporting the mental health of vulnerable groups becomes especially challenging. Video interventions have also been shown to modify health beliefs and effectively inform public health measures, particularly those attitudes and practices toward COVID-19 prevention. For example, Elasrag et al. [3] designed a video intervention showing a COVID-informative video to 50 middle-aged nurses. Before the intervention, 70% of the had poor knowledge about COVID-19 prevention. After the video intervention, 76% had good knowledge and only 6% had poor knowledge. Moreover, after the introduction of an educational program, over 90% displayed positive effects on their understanding and attitude, and practice.

Another video intervention in Nigeria geared towards young researchers unveiled significant improvements in COVID-19 knowledge [20]. In that study, authors stated the need for regular online educational programs to enhance COVID-19 understanding. The same strategy (a video intervention on COVID) was conducted in Israel to 501 participants [10]. This study found that COVID-related knowledge, perceived knowledge, perceived safety, and resilience increased significantly after the educational program intervention to varying degrees.

In this study, we introduce a new iteration of a method to assess the effectiveness of video interventions delivered through WhatsApp. This approach aimed to reach individuals with limited resources in Ecuador, a country marked by social inequalities. The goal was to enhance their understanding of crucial COVID-19 information, especially given the significant excess death rate the nation experienced during the pandemic [21–24]. To preview our findings, after watching an informative video, participants exhibited improved understanding of COVID-19. Specifically, they recognized that individuals over 65 years old are the most vulnerable group, gained a heightened awareness of the virus's incubation period, and notably dispelled the naive belief in an effective antidote for COVID-19.

## Materials and methods

### Recruitment and short video intervention

Participants were randomly selected from a dataset of COVID-19 patients in Ecuador. We sent pre- and post-intervention questionnaires and videos intervention to the 87 patients who agreed to participate in the study through WhatsApp. First, they replied to a questionnaire and then watched the video intervention. Finally, they replied to the same questionnaire. All participants were surveyed in June 2021. The intervention consisted of a short video of 1:30 minutes, targeted at the adult population to impact their knowledge and understanding of the COVID-19 pandemic and give new tools to address this developing sanitary crisis. The content encompassed key aspects of the virus, primary preventive measures, vulnerable age brackets, and typical symptoms. The information in the video was delivered through a YouTube URL via a WhatsApp message and a series of instructions to fill out the pre-intervention and post-intervention surveys were attached.

### Questionnaire

Considering the importance of the COVID-19 pandemic [25], the study was conducted using a series of questions based on multiple-choice answers. The questionnaire was composed of brief greetings and an introduction to its objective through background information, goals, and procedures, and emphasizing the voluntary nature of participation.

The survey included eight demographic questions to characterize the participants. It inquired about their gender, providing choices of either female or male. Additionally, participants were prompted to specify their type of residence, choosing among urban, semi-urban, or rural locations. For marital status, they could select from single, married, or cohabiting. Regarding education level, categories ranged from those who completed junior high school to those with postgraduate degrees, including master's and PhDs. Participants were also given the option to disclose their average annual income, with choices ranging from less than $9,999 to $99,999; however, there was an additional option allowing them to decline answering. The section concluded with a question on occupational activity, categorizing respondents into sectors such as the private sector, public sector, student, retiree, or unpaid job.

Subsequently, the survey presented a set of 30 multiple-choice questions that gauged participants' perceptions regarding the government's handling of the pandemic, their personal understanding of the virus's mortality rate, incubation period, and transmissibility, their beliefs about the existence of an effective cure, the efficacy of mask-wearing in preventing infection, and the effectiveness of quarantine measures. Of these questions, 19 had straightforward response options: either 'yes-no' or a 'yes-no-maybe' format. The remaining questions offered an extended multiple-choice answer grid.

Recognizing the importance of instrument reliability and validity and given the immediate need for data collection during the early stages of the pandemic, a rapid validation process was undertaken. The questionnaire was reviewed and validated by a panel of five experts specializing in constructing health-related questionnaires. Their feedback helped refine the questions, ensuring they were clear, relevant, and appropriate for the intended audience. To further assess the instrument's reliability, we piloted the questionnaire with 10 individuals. Their responses and feedback provided insights into the questionnaire's clarity, comprehensibility, and relevance, leading to final adjustments before widespread dissemination.

This study was performed in compliance with the World Medical Association Declaration of Helsinki on Ethicals and was approved by the ethics committee: Comité de ética e Investigación en Seres Humanos (CEISH), Guayaquil-Ecuador (#HCK-CEISH-18-0060). Informed

consent was obtained from all the subjects involved in the study. Written informed consent was obtained from all patients publish this paper.

## Statistical analysis

Descriptive statistics were used to outline the primary demographic characteristics of the participants. Subsequently, both pre- and post-intervention descriptive statistics were calculated for the variables of interest. Since our study involved the same participants pre- and post-intervention (indicating dependent samples), the McNemar test was initially considered. However, given that the majority of our variables were not dichotomous, we opted for the marginal homogeneity chi-square test to assess the presence of statistically significant shifts in proportions after the intervention [26–28]. P-values less than 0.05 were denoted with a "*" and deemed significant, while those between 0.05 and 0.1 were marked with a "+" and considered marginally significant. Values above 0.1 were viewed as non-significant. All statistical analyses were conducted using Stata 17.0 [26].

## Results

The study evaluated 87 participants before and after observing a short informative video about COVID-19. Table 1 shows the demographic variables. The sample comprised 71.26%, women, and 28.74%, men. Respondents were distributed in three main areas: urban (68.97%), semi-urban (14.94%), and rural (16.09%). The study group included individuals with varied marital statuses: 63.22% were single, 34.48% were married, and 2.30% were cohabiting. Their educational backgrounds were also diverse: 2.3% had completed junior high school, 18.39% had finished high school, 64.37% had a bachelor's or university degree, and 14.94% had pursued postgraduate studies. Participants were also given an optional question about their income level. Most, 37.93%, reported an income under $9,999 per year. In terms of employment status, 28.74% indicated they worked in the private sector, 16.09% in the public sector, 37.93% identified as students, 2.30% were retirees, and 14.94% were engaged in unpaid jobs.

Table 2 showcases the findings from 30 distinct opinion-based questions that assessed participants' knowledge about COVID-19. The P-value is derived from the Stuart-Maxwell Homogeneity test, with significance levels indicated as: P < 0.05 (*) and P < 0.1 (+). Eight of these questions revealed significant shifts in answers after the respondents watched an informative video about COVID-19.

For instance, after viewing the video, there was a 15.92% increase in participants who identified those over 65 years of age as the group most vulnerable to the disease, achieving a significance level of P = 0.06. When questioned about the virus's incubation period, the number of respondents believing it to be 1–7 days dropped by 27.58%, while those acknowledging an incubation period of 1–14 days rose by 28.74% (P = 0.00).

Furthermore, there was a 10.34% increase in participants who understood that COVID-19 is contagious during its incubation period (P = 0.01). The belief in the existence of an effective cure for COVID-19 dropped by 16.09% post-video, with a notable 20.69% increase among those who hadn't viewed the video (P < 0.00).

In gauging participants' subjective opinions on the successful control of the COVID-19 pandemic, 31.03% believed it was being controlled effectively, 40.23% denied the existence of the pandemic, and 28.74% remained undecided. After viewing the video, these percentages shifted to 36.78%, 47.13%, and 16.09% respectively, though the change wasn't statistically significant (P > 0.05).

The belief in vaccines as a preventative measure against contracting the disease also saw a shift. Initially, only 14.94% of participants believed vaccines could prevent them from getting

**Table 1. Characteristics of the study population (n = 87).**

| Variables | |
|---|---|
| *Gender* | |
| Female | 71.26% |
| Male | 28.74% |
| *Area* | |
| Urban | 68.97% |
| Semi-urban | 14.94% |
| Rural | 16.09% |
| *Marital status* | |
| Single | 63.22% |
| Married | 29.89% |
| Cohabiting | 2.30% |
| Divorced | 4.60% |
| *Level of Education* | |
| Completed junior high school | 2.30% |
| Completed high school | 18.39% |
| Bachelor/University Student | 64.37% |
| Postgraduate Studies/Master/PhD | 14.94% |
| *Income* | |
| $10,000-$49,999/year | 17.24% |
| $100,000-$149,999/year | 1.15% |
| $50,000-$99,999/year | 2.30% |
| ≤$9,999/year | 41.38% |
| Do not answer | 37.93% |
| *Profession* | |
| Private sector | 28.74% |
| Public sector | 16.09% |
| Student | 37.93% |
| Retiree | 2.30% |
| Unpaid jobs | 14.94% |

the disease. This belief increased to 25.29% after watching the video, with a significant P-value of 0.02. Furthermore, the question assessing the belief that the COVID-19 vaccine could prevent viral transmission revealed the most significant change after participants watched the video, with a 10.35% increase in affirmative responses.

Finally, after viewing the video, there was an 8.04% increase in participants who believed that the COVID vaccine could protect them from the delta variant. Conversely, there was a 3.44% rise in those who believed it couldn't, and an 11.5% decrease in those who were unsure or believed it might offer protection.

On the other hand, even if the remaining questions were not statistically significant, it is crucial to note the behavioral shifts they revealed post-intervention within the sample. Among the five most relevant non-significant questions, there was a 1.15% shift in respondents who initially felt that children and young adults did not need to take specific actions, moving to a more uncertain stance (e.g., 'I do not know') after the video intervention.

Similarly, there was a modest 2.3% rise in individuals who, initially uncertain about airborne transmission of COVID-19, became convinced of its possibility after watching the video. Additionally, there was a 1.15% increase in those who came to believe that respiratory microdroplets could indeed spread COVID-19 after initially thinking otherwise.

**Table 2. Percentages of knowledge, attitude, and practice of COVID-19 between T1 and T2 (pre-post intervention) (n = 87).**

| Variables | T1 | T2 | Prob>$\chi^2$ |
|---|---|---|---|
| Do you agree that COVID-19 is as deadly as seasonal flu? | | | |
| Yes | 54.02% | 56.32% | 0.37 |
| No | 41.38% | 41.38% | |
| I do not know | 4.60% | 2.30% | |
| Do you agree that COVID-19 can be prevented with the flu vaccine? | | | |
| Yes | 21.84% | 18.39% | 0.14 |
| No | 67.82% | 78.16% | |
| I do not know | 10.34% | 3.45% | |
| Which age group do you think is most susceptible to COVID-19?[1] | | | |
| 0–2 years old | 16.54% | 18% | 0.06+ |
| 2–18 years old | 17.32% | 12.07% | |
| 18–40 years old | 21.26% | 18.97% | |
| 40–65 years old | 31.50% | 22% | |
| >65 years old | 13.39% | 29.31% | |
| What do you think is the incubation period of COVID-19? | | | |
| 1–14 days | 60.92% | 89.66% | 0.00* |
| 1–7 days | 36.78% | 9.20% | |
| I do not know | 1.15% | 1.15% | |
| Supposedly at 4 days of contact the first symptoms are seen | 1.15% | 0% | |
| Do you think COVID-19 is contagious during the incubation period? | | | |
| Yes | 88.51% | 98.85% | 0.01* |
| No | 8.05% | 1.15% | |
| I do not know | 3.45% | 0% | |
| Do you think there is an effective cure for COVID-19? | | | |
| Yes | 31.03% | 14.94% | 0.00* |
| No | 58.62% | 79.31% | |
| I do not know | 10.34% | 5.75% | |
| Do you think wearing a mask is necessary to prevent infection? | | | |
| Yes | 98.85% | 100% | 0.32 |
| No | 1.15% | 0% | |
| Do you think children and young adults need to take necessary action? | | | |
| Yes | 98.85% | 98.85% | 0.37 |
| No | 1.15% | 0% | |
| I do not know | 0% | 1.15% | |
| Do you think COVID-19 is spread through aero transmission? | | | |
| Yes | 78.16% | 80.46% | 0.35 |
| No | 18.39% | 18.39% | |
| I do not know | 3.45 | 1.15 | |
| Do you think COVID-19 is spread by respiratory microdroplets? | | | |
| Yes | 97.70% | 98.85% | 0.56 |
| No | 2.30% | 1.15% | |
| Do you think you should quarantine for 14 days if you have been exposed? | | | |
| Quite disagreeing | 10.34% | 14.94% | 0.22 |
| Disagreeing | 4.60% | 3.45% | |
| Neither agree nor disagree | 6.90% | 2.30% | |
| I agree | 36.78% | 35.63% | |
| Quite agree | 41.38% | 43.68% | |

*(Continued)*

**Table 2.** (Continued)

| Variables | T1 | T2 | Prob>$\chi^2$ |
|---|---|---|---|
| How worried are you about COVID-19? | | | |
| Nothing worried | 3.45% | 3.45% | 0.88 |
| Minimally worried | 12.64% | 12.64% | |
| Moderately concerned | 49.43% | 51.72% | |
| Very worried | 34.48% | 32.18% | |
| How frequently do you check for updates ON COVID-19? | | | |
| Less than once a day | 43.68% | 39.08% | 0.14 |
| Once a day | 44.83% | 48.28% | |
| More than once a day | 11.49% | 12.64% | |
| What impact the COVID-19 pandemic has had on your life? | | | |
| None | 0% | 1.15% | 0.42 |
| Minimal | 11.49% | 8.05% | |
| Moderate | 44.83% | 48.28% | |
| Significant | 43.68% | 42.53% | |
| How do you feel about the way the COVID-19 pandemic is being handled? | | | |
| Very pessimistic | 4.60% | 4.60% | 0.98 |
| Moderately pessimistic | 12.64% | 13.79% | |
| Neutral | 33.33% | 32.18% | |
| Moderately optimistic | 31.03% | 29.89% | |
| Very optimistic | 18.39% | 19.54% | |
| Do you think the COVID-19 pandemic will be successfully controlled soon? | | | |
| Yes | 31.03% | 36.78% | 0.02* |
| No | 40.23% | 47.13% | |
| I do not know | 28.74% | 16.09% | |
| How often have you visited places where there was crowding of people? | | | |
| Never | 12.64% | 13.79% | 0.40 |
| Rarely | 58.62% | 60.92% | |
| Often | 28.74% | 25.29% | |
| Have you recently used a mask when you go out? | | | |
| Often | 16.09% | 17.24% | 0.32 |
| Always | 83.91% | 82.76% | |
| Would you like to see an educational video or brochure about COVID-19? | | | |
| Yes | 82.76% | 0% | - |
| No | 12.64% | 0% | |
| I do not know | 4.60% | 0% | |
| N/A | 0% | 100% | |
| For you, what would be a good duration for the video? | | | |
| 1–2 minutes | 62.07% | 0% | - |
| Less than 1 minute | 14.94% | 0% | |
| More than 3 minutes | 22.99% | 0% | |
| N/A | 0% | 100% | |
| Have you been diagnosed with COVID-19? | | | |
| Yes | 40.23% | 40.23% | 1.00 |
| No | 59.77% | 59.77% | |
| Has any family member/acquaintance of yours tested positive for COVID 19? | | | |
| Yes | 85.06% | 85.06% | 1.00 |
| No | 14.94% | 14.94% | |
| Would you like to have an application on your cell phone to be updated? | | | |

*(Continued)*

**Table 2.** (Continued)

| Variables | T1 | T2 | Prob>$\chi^2$ |
|---|---|---|---|
| Yes | 62.07% | 64.37% | 0.26 |
| No | 31.03% | 28.74% | |
| I do not know | 6.90% | 6.90% | |
| Have you been vaccinated against COVID-19? | | | |
| Yes | 95.40% | 95.40% | 1.00 |
| No | 4.60% | 4.60% | |
| If you have been vaccinated, which vaccine have you received? | | | |
| AstraZeneca | 18.39% | 18.39% | 1.00 |
| Janssen | 2.30% | 2.30% | |
| Modern | 3.45% | 3.45% | |
| I have not been vaccinated | 3.45% | 3.45% | |
| Pfizer-BioNTech | 40.23% | 40.23% | |
| Sinovac | 32.18% | 32.18% | |
| Do you believe COVID-19 vaccine can prevent you from getting it? | | | |
| Yes | 14.94% | 25.29% | 0.02* |
| No | 68.97% | 58.62% | |
| Maybe | 16.09% | 16.09% | |
| Do you believe the COVID-19 vaccine can prevent you from transmitting the virus? | | | |
| Yes | 16.09% | 26.44% | 0.02* |
| No | 63.22% | 59.77% | |
| Maybe | 20.69% | 13.79% | |
| Have you heard of the Delta variant of COVID-19? | | | |
| Yes | 98.85% | 97.70% | 0.61 |
| No | 0% | 1.15% | |
| Maybe | 1.15% | 1.15% | |
| Do you believe the COVID-19 vaccine can protect you from the Delta variant? | | | |
| Yes | 27.59% | 35.63% | 0.07+ |
| No | 29.89% | 33.33% | |
| Maybe | 42.53% | 31.03% | |
| Do you the Delta variant of COVID-19 is more lethal? | | | |
| Yes | 51.72% | 57.47% | 0.34 |
| No | 20.69% | 21.84% | |
| Maybe | 27.59% | 20.69% | |

[1] Participants could chose more than one (1) answer. Results may add up to more than 100%

Concerning quarantine durations post-exposure, the percentage of participants who felt that a 14-day quarantine was not necessary increased from 10.34% to 14.94%, suggesting an unintended consequence of the video. There was a slight uptick, from 41.38% to 43.68%, in respondents who endorsed the 14-day quarantine. Lastly, the video seemed to have a positive influence on mask-wearing habits: there was a 1.15% increase in participants who reported more frequent mask use when going out after viewing the video.

## Discussion

After analyzing the collected data from a pool of varied opinion questions about participants' knowledge of COVID-19, we found eight, which presented significant results.

After viewing the educational video, participants demonstrated increased awareness of the age group most susceptible to the disease. Additionally, their understanding of the COVID-19 incubation period expanded, with over a quarter more respondents aligning with the scientifically accepted range of 1 to 14 days. There was also a notable rise in individuals who recognized that COVID-19 is contagious during its incubation, underscoring enhanced knowledge about the virus's behavior and transmissibility.

Among the most significant findings of our study, eight stood out not solely due to their statistical significance but also because of the profound societal implications they hold, particularly the decline in the naive belief of an existing effective antidote for COVID-19. These insights complement findings from other research, such as that by Kaim et al. [10], which highlighted the efficacy of educational interventions delivered through brief videos. Such a format offers a feasible and potent means of informing the public and should be viewed as a vital tool in handling potential health crises.

Furthermore, the impact of educational video interventions on nurses' knowledge, attitudes, and practices can be likened to our findings. In a study by Elasrag in Egypt [3], prior to the intervention, 70% of participants displayed poor knowledge. Post-intervention, however, 76% demonstrated good knowledge, reflecting a growth of 6 percentage points in knowledge. In our study, 30% of participants initially had limited knowledge, which improved by 15% following the video treatment. The differences in effect size may stem from variations in the samples between the two studies.

While the magnitudes of change between our study and Elasrag's might not seem directly comparable, the consistent direction of change across both researches strengthens our findings. Given the similarity in the effects observed across various questions, our results could potentially be applied to Ecuador's neighboring countries in Latin America. This region, deeply impacted by the pandemic, grapples with diverse healthcare inequalities [6, 19, 21, 29]. Our findings remain significant as they underscore the influence of brief video interventions on the social behavior of a population segment, reinforcing the potential of such interventions [30].

One of the limitations of this study is that participants' behavior was observed over a single period. This approach overlooks potential interactions with unaccounted variables over time and the long-term impact of learning through audiovisual materials. Moreover, the research relied on a convenience sample, which, due to its relatively small size, restricts the broad applicability of the findings. However, given that the respondent pool was predominantly middle-class individuals, there's potential to generalize the results more broadly within the non-medical segment of the population.

In line with Chan et al. [13], we contend that additional research is needed to examine the impact of educational interventions on a larger cohort, particularly focusing on those outside the medical sphere of COVID-19. Such research would allow for a more precise understanding of the intervention's magnitude and direction, thereby guiding the formulation of more effective social policies in the event of a crisis.

## Conclusion

Our study underscores the transformative power of targeted interventions in influencing the public's knowledge, attitudes, and practices (KAP) about COVID-19. Notably, out of 28 areas assessed, 8 exhibited significant shifts post-intervention, illustrating the effectiveness of our approach.

Moreover, several other areas exhibited encouraging trends, pointing towards potential focal points for future interventions or studies. The implications of these findings are

profound. They highlight the potential of a brief and low-cost intervention to not only educate but also empower the public amidst health crises like the COVID-19 outbreak.

Considering the context of our study in Ecuador, the results might offer insights that could be relevant to other settings with similar characteristics. Many Latin American populations, akin to the Ecuadorian populace, were unacquainted with the challenges and implications of a pandemic–from lockdowns to quarantines. While our sample was limited, and generalization should be approached with caution, the findings do suggest potential avenues for exploration in broader contexts.

Looking ahead, the lasting impact of these educational interventions on individuals' capacities to modify their knowledge, adopt preventive measures, and change behaviors necessitates further exploration. Such research endeavors will be instrumental in fortifying our preparedness and response to future outbreaks.

## Supporting information

**S1 Dataset.**
(ZIP)

## Acknowledgments

We thank Alejandro Curbelo for his valuable assistance during the project.

## Author Contributions

**Conceptualization:** Genesis Camacho-Leon, Roberto Páez-Plúas, Azza Sarfraz, Jack Michel, Ivan Cherrez-Ojeda.

**Data curation:** Roberto Páez-Plúas.

**Formal analysis:** Roberto Páez-Plúas.

**Funding acquisition:** Ivan Cherrez-Ojeda.

**Investigation:** Genesis Camacho-Leon, Azza Sarfraz, Zouina Sarfraz, Jack Michel.

**Methodology:** Genesis Camacho-Leon, Azza Sarfraz, Zouina Sarfraz.

**Writing – original draft:** Marco Faytong-Haro, Roberto Páez-Plúas.

**Writing – review & editing:** Marco Faytong-Haro, Roberto Páez-Plúas, Azza Sarfraz, Zouina Sarfraz, Jack Michel, Ivan Cherrez-Ojeda.

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
