## [Decision Letter · Decision Letter 0]

18 Aug 2023

PONE-D-23-22429The Influence of a Quick Educational Video Intervention on COVID-19-related Knowledge in EcuadorPLOS ONE

Dear Dr. Cherrez-Ojeda,

Thank you for submitting your manuscript to PLOS ONE. After careful consideration, we feel that it has merit but does not fully meet PLOS ONE’s publication criteria as it currently stands. Therefore, we invite you to submit a revised version of the manuscript that addresses the points raised during the review process.

Please submit your revised manuscript within one week. If you will need more time than this to complete your revisions, please reply to this message or contact the journal office at plosone@plos.org. Please include the following items when submitting your revised manuscript:A rebuttal letter that responds to each point raised by the academic editor and reviewer(s). You should upload this letter as a separate file labeled 'Response to Reviewers'.A marked-up copy of your manuscript that highlights changes made to the original version. You should upload this as a separate file labeled 'Revised Manuscript with Track Changes'.An unmarked version of your revised paper without tracked changes. You should upload this as a separate file labeled 'Manuscript'.If applicable, we recommend that you deposit your laboratory protocols in protocols.io to enhance the reproducibility of your results. Protocols.io assigns your protocol its own identifier (DOI) so that it can be cited independently in the future. For instructions see: https://journals.plos.org/plosone/s/submission-guidelines#loc-laboratory-protocols. Additionally, PLOS ONE offers an option for publishing peer-reviewed Lab Protocol articles, which describe protocols hosted on protocols.io. Read more information on sharing protocols at https://plos.org/protocols?utm_medium=editorial-email&utm_source=authorletters&utm_campaign=protocols.

We look forward to receiving your revised manuscript.

Kind regards,

Nour Amin Elsahoryi, pHD

Academic Editor

PLOS ONE

Journal Requirements:

"This research was funded by Universidad Espíritu Santo, grant number 2021-MED-001."

Reviewers' comments:

Reviewer's Responses to Questions

**Comments to the Author**

1. Is the manuscript technically sound, and do the data support the conclusions?

Reviewer #1: No

Reviewer #2: Yes

2. Has the statistical analysis been performed appropriately and rigorously? 

Reviewer #1: No

Reviewer #2: Yes

3. Have the authors made all data underlying the findings in their manuscript fully available?

Reviewer #1: Yes

Reviewer #2: Yes

4. Is the manuscript presented in an intelligible fashion and written in standard English?

Reviewer #1: No

Reviewer #2: Yes

5. Review Comments to the Author

Reviewer #1: Firs t I would like to thank the authors for their effort. But I have some major comments

Major comments.

1. Conclusion is not supported by the statistical outputs. For example out of 28 chi square test only 6 are significant.

2. However chi square test do not support "This study

shows that educational intervention positively impacts the knowledge of the people who

watch it." in the conclusion.

3. How did you validate and assess the reliability of the questioner.It is very import.

4. Small sample size does not show any possibility of generalizing the tool practice in mass scale.

5. Chi square test is not the ideal test to show per and post intervention effect. If can develop pre and post aggregated score like mean values the can use paired sample test for the purpose.

6.. Regression analysis will benefited to adjust confounding effect.

7. Table one is not very user friendly. Need readjust.

8. I nave not seen reference list in the manuscript.

8. Abstract need more statistical values to shoe the results.

Reviewer #2: Interesting, well written and well designed review The Influence of a Quick Educational Video Intervention on COVID-19-related Knowledge and authors have made great efforts in doing this paper

Overall a relook at the grammar and rephrasing of certain sentences is advised.

6. PLOS authors have the option to publish the peer review history of their article (what does this mean?). If published, this will include your full peer review and any attached files.

Reviewer #1: No

Reviewer #2: No

---

## [Author Response · Author response to Decision Letter 0]

29 Aug 2023

We have submitted a rebuttal letter addressing all the comments from reviewers.

---

## [Editor Report · Decision Letter 1]

6 Sep 2023

The Influence of a Quick Educational Video Intervention on COVID-19-related Knowledge in Ecuador

PONE-D-23-22429R1

Dear Dr. 

We’re pleased to inform you that your manuscript has been judged scientifically suitable for publication and will be formally accepted for publication once it meets all outstanding technical requirements.

Kind regards,

Nour Amin Elsahoryi, pHD

Academic Editor

PLOS ONE

---

## [Editor Report · Acceptance letter]

27 Sep 2023

PONE-D-23-22429R1 

The Influence of a Quick Educational Video Intervention on COVID-19-related Knowledge in Ecuador 

Dear Dr. Cherrez-Ojeda:

I'm pleased to inform you that your manuscript has been deemed suitable for publication in PLOS ONE. Congratulations! Your manuscript is now with our production department. 

Kind regards, 

on behalf of

Dr. Nour Amin Elsahoryi 

Academic Editor

PLOS ONE